# Learning from Positive and Unlabeled Data with a Selection Bias

Masahiro Kato[1,2], Takeshi Teshima[1,2], and Junya Honda[1,2]

[1]The University of Tokyo, Tokyo, Japan
[2]RIKEN, Tokyo, Japan
*{mkato, teshima}@ms.k.u-tokyo.ac.jp, honda@edu.k.u-tokyo.ac.jp*

## Abstract

We consider the problem of learning a binary classifier only from *positive* data and *unlabeled* data (*PU learning*). Recent methods of PU learning commonly assume that the labeled positive data are identically distributed as the unlabeled positive data. However, this assumption is unrealistic in many instances of PU learning because it fails to capture the existence of a *selection bias* in the labeling process. When the data has selection bias, it is difficult to learn the Bayes optimal classifier by conventional methods of PU learning. In this paper, we propose a method to *partially identify* the classifier. The proposed algorithm learns a scoring function that preserves the order induced by the class posterior under mild assumptions, which can be used as a classifier by setting an appropriate threshold. Through experiments, we show that the method outperforms previous methods for PU learning on various real-world datasets.

## 1 Introduction

We consider a situation that there are only *positive* and *unlabeled* data, and train a binary classifier only from them (*PU learning*). This problem arises in various practical situations, such as information retrieval and outlier detection (Elkan & Noto, 2008; Ward et al., 2009; Scott & Blanchard, 2009; Blanchard et al., 2010; Li et al., 2009; Nguyen et al., 2011). One of the milestones of PU learning is Elkan & Noto (2008), who proposed a practically useful algorithm with theoretical analysis, and there is subsequent research called *unbiased PU learning* (du Plessis & Sugiyama, 2014; du Plessis et al., 2015) where an *unbiased estimator* of the classification risk is minimized.

We focus on the *case-control* scenario (a.k.a. the *problem setting based on two samples of data*; Ward et al., 2009; Niu et al., 2016). In this scenario, positive data are obtained separately from unlabeled data, and unlabeled data are sampled from the whole population. As Elkan & Noto (2008) explained, we cannot identify a classifier without an assumption on how positive data are labeled. Therefore "*selected completely at random*" (SCAR) is traditionally assumed, i.e., the positive labeled data are identically distributed as the positive unlabeled data (Elkan & Noto, 2008; du Plessis et al., 2015). The assumption of SCAR is, however, unrealistic in many instances of PU learning, e.g., a patient's electronic health record (Bekker & Davis, 2018a) and a recommendation system (Marlin & Zemel, 2009; Schnabel et al., 2016). In these cases, there is a *selection bias* (Heckman, 1979; Manski, 2008; Angrist & Pischke, 2008); the distribution of the positive data may differ between the labeled data and the unlabeled data.

Several recent related works have proposed alternative assumptions (Bekker & Davis, 2018a;b). However, in order to weaken SCAR, they impose other additional assumptions. In this work, we consider a more natural assumption such that SCAR becomes its special case. We assume that $p(o = +1|\boldsymbol{x}, y = +1)$ and $p(y = +1|\boldsymbol{x})$ induce the same ordering on the input space $\mathcal{X}$, where $y \in \{-1, +1\}$ is the data label and $o = +1$ (resp. $o = 0$) denotes the event that the data is observed (resp. not observed). We call this property the *invariance of order*. In the real-world application, there are many situations with a selection bias which follows the invariance of order. Among them, we list the following two examples.

**Example 1:**   (Anomaly Detection)
The goal of *anomaly detection* is to find anomaly data in an unlabeled dataset based on another dataset that consists only of anomaly data. When the anomaly data is collected, the more likely a datum is an anomaly, the more likely it is noticed and gets labeled.

**Example 2:**   (Face Recognition)
The goal of this task is to identify a user from a set of face images based on some pictures of the user. In this case, the positive data are the face images identified with the user, and the unlabeled data consist of all the unidentified face images. The user may more likely provide pictures in which the face can be seen clearly, while in the unlabeled data there may be many unclear images.

Our problem setting is similar to the problem called *learning from instance-dependent noisy labels* (Du & Cai, 2015; Bootkrajang, 2016). In this problem setting, positive and negative data are available, but the labels are subject to the noise that flips the label with instance-dependent probability. Besides, there are existing works putting assumption similar to the invariance order (Du & Cai, 2015; Bootkrajang, 2016). We explain the difference between their works and ours around Assumption 1 in Section 2.1. We name our problem setting *PU learning with a Selection Bias* (PUSB).

In this paper, we propose a novel framework to deal with this problem setting. The experimental results show that our proposed method is appropriate for real-world applications compared to existing approaches for PU learning.

## 2   PROBLEM SETTING OF PU LEARNING WITH A SELECTION BIAS

We consider a binary classification problem to classify $x \in \mathcal{X} \subset \mathbb{R}^d$ into one of the two classes $\{-1, +1\}$. We assume that there exists a joint distribution $p(x, y, o)$, where $y \in \{-1, +1\}$ is the class label of $x$, and $o \in \{0, +1\}$ is the observation status of $y$ (observed if $o = +1$ and unobserved if $o = 0$). In other words, $x$ is labeled if $o = +1$, and it is unlabeled otherwise.

In PU learning, there are two distinguished sampling schemes called *one sample* and *two samples* of data (Niu et al., 2016). They are also called the *censoring scenario* and *case-control scenario*, respectively (Elkan & Noto, 2008). In the censoring scenario, a set of unlabeled data is sampled from the marginal density $p(x)$. Then, if a data point $x$ is positive, it gets labeled with probability $p(o = +1|x, y = +1)$; if $x$ is negative, it is never labeled. In the case-control scenario, a set of positive data is drawn from the positive conditional density $p(x|y = +1)$ and a set of unlabeled data is drawn from $p(x)$. As Niu et al. (2016) stated, the case-control scenario is slightly more general than the censoring scenario setting. It is because the censoring scenario assumes the access to samples generated by $p(x)$, $p(x|o = +1)$, and $p(x|o = 0)$ whereas the case-control scenario only assumes the access to samples generated by $p(x)$ and $p(x|o = +1)$. Therefore we focus on the case-control scenario.

Suppose that we have a positive dataset $\{x_i\}_{i=1}^n$ and an unlabeled dataset $\{x_i'\}_{i=1}^{n'}$

$$\{x_i\}_{i=1}^n \overset{\text{i.i.d.}}{\sim} p(x|y = +1, o = +1), \ \{x_i'\}_{i=1}^{n'} \overset{\text{i.i.d.}}{\sim} p(x).$$

We assume that negative data are never labeled.

Note that we do not assume SCAR. Therefore, $p(x|y = +1)$ may differ from $p(x|y = +1, o = +1)$. In the case they differ, we say that there is a *selection bias*.

The quantity $\pi = p(y = +1)$ is called the *class-prior*. In our problem setting, we assume that $\pi$ is known. For example, in anomaly detection, the percentage of anomaly in the whole batch of products can be reported based on past experiences. Although there are various methods for estimating the class-prior in the traditional framework of the case-control scenario (du Plessis et al., 2016; Ramaswamy et al., 2016; Jain et al., 2016; Kato et al., 2018), we cannot estimate the class-prior in our problem setting under a theoretical guarantee. In Section 5, we show how misspecified class priors affect the performance of a classifier. As we explain later, even if we do not know the class-prior, we only have to change the last step of our algorithm. In summary, our goal is to obtain a classifier $h : \mathcal{X} \to \{-1, 1\}$ only from $\{x_i\}_{i=1}^n$, $\{x_i'\}_{i=1}^{n'}$, and $\pi$ under a weaker assumption than SCAR.

## 2.1 IDENTIFICATION STRATEGY

As stated by Elkan & Noto (2008), even if the class prior is given, we cannot estimate $p(y = +1|\boldsymbol{x})$ only from positive data and unlabeled data without any assumption in PU learning. In the case-control scenario, a standard assumption is SCAR, i.e. $p(\boldsymbol{x}|y = +1, o = +1) = p(\boldsymbol{x}|y = +1, o = 0)$, so that $p(y = +1|\boldsymbol{x})$ can be estimated from the data in principle. However, in many instances of PU learning, the SCAR assumption is unreasonable as discussed before. Therefore, we relax SCAR and accommodate a selection bias. We can see how SCAR makes the class posterior identifiable in the following equation:

$$p(y = +1|\boldsymbol{x}) = \frac{p(\boldsymbol{x}, y = +1)}{p(\boldsymbol{x})} = \frac{p(\boldsymbol{x}|y = +1)\pi}{p(\boldsymbol{x})} \underbrace{=}_{\text{SCAR}} \frac{p(\boldsymbol{x}|y = +1, o = +1)\pi}{p(\boldsymbol{x})}.$$

Here, $p(\boldsymbol{x}|y = +1, o = +1)$ can be estimated from the sample. This estimate can be used to obtain an estimate of $p(\boldsymbol{x}|y = +1)$ and hence that of $p(y = +1|\boldsymbol{x})$ if we assume SCAR. However, without assuming SCAR, $p(\boldsymbol{x}|y = +1)$ may differ from $p(\boldsymbol{x}|y = +1, o = +1)$, and $p(y = +1|\boldsymbol{x})$ is not identifiable.

Therefore, instead of estimating $p(y = +1|\boldsymbol{x})$, we consider extracting some useful information of $p(y = +1|\boldsymbol{x})$ to learn a classifier. This kind of approach is known as "partial identification" (Manski, 2008) in statistics and economics. First, we introduce an assumption that is weaker than SCAR.

**Assumption 1** (Invariance of Order). *For any $\boldsymbol{x}_i, \boldsymbol{x}_j \in \mathcal{X}$, we have*

$$p(y = +1|\boldsymbol{x}_i) \leq p(y = +1|\boldsymbol{x}_j) \Leftrightarrow p(o = +1|\boldsymbol{x}_i) \leq p(o = +1|\boldsymbol{x}_j).$$

Although Assumption 1 does not allow one to construct an unbiased estimator of the risk functional, we try to partially identify $p(y = +1|\boldsymbol{x})$ under this assumption. Our problem setting can be regarded as a generalization of the traditional case-control scenario because SCAR is a special case of the invariance of order.

## 2.2 RELATED WORKS

A similar assumption can be found in the literature of learning from instance-dependent noisy labels (Du & Cai, 2015; Bootkrajang, 2016), which considers a probabilistic label flipping that is proportional to $p(y = +1|\boldsymbol{x})$. However, in order to apply methods of learning from noisy labels to PU learning, we need to assume the censoring scenario and these methods cannot be applied to our problem setting based on the case-control scenario. The censoring scenario is a special case of learning from noisy labels where only negative data is contaminated, i.e., some positive labels flip to negative labels. Thus, in the censoring scenario, unlabeled data can be regarded as negative-labeled data contaminated by positive data. On the other hand, in the case-control scenario, unlabeled data is generated from the marginal distribution $p(\boldsymbol{x})$, i.e., we cannot observe samples generated from $p(\boldsymbol{x}|o = 0)$. Therefore, our problem setting, namely the case-control scenario with invariance of order, is different from the existing works of learning from instance-dependent noisy labels. In addition, our method is also applicable to the censoring scenario when the invariance of order holds because the unlabeled data of the case-control scenario can be made from positive and unlabeled data of the censoring scenario.

In Example 1, $p(y = +1|\boldsymbol{x})$ is the probability that a given input $\boldsymbol{x}$ is anomaly, while $p(o = +1|\boldsymbol{x})$ is the probability that a given input $\boldsymbol{x}$ gets labeled in the dataset. In Example 2, a positively labeled data is an image $\boldsymbol{x}$ that is known to belong to a user. Here, $p(o = +1|\boldsymbol{x})$ is the probability that the user provides the picture $\boldsymbol{x}$ as a training datum.

## 3 STRATEGY FOR PARTIAL IDENTIFICATION AND CLASSIFICATION

As discussed in Section 2.1, we cannot estimate $p(y = +1|\boldsymbol{x})$ when there is a selection bias even if the class prior is given. Our idea of partial identification is based on the following theorem with the density ratio

$$r(\boldsymbol{x}) = \frac{p(\boldsymbol{x}|y = +1, o = +1)}{p(\boldsymbol{x})}. \tag{1}$$

---

**Algorithm 1** Conceptual Algorithm in Population

---

**Input:** $p(\boldsymbol{x}|y = +1)$, $p(\boldsymbol{x})$ and the class-pror $\pi$.
Using $p(\boldsymbol{x}|y = +1)$ and $p(\boldsymbol{x})$, calculate $r(\boldsymbol{x})$ by minimization of either (4) or (7).
Using $r(\boldsymbol{x})$, calculate $\theta_\pi$ in (2).
Using the density ratio $r(\boldsymbol{x})$ and the threshold $\theta_\pi$, obtain a classifier $h(\boldsymbol{x}) = \mathtt{sign}(r(\boldsymbol{x}) - \theta_\pi)$.

---

**Theorem 1** (Order Preserving Property of the score Function). *Suppose that Assumption 1 holds. Then, for any $\boldsymbol{x}_i, \boldsymbol{x}_j \in \mathcal{X}$,*

$$p(y = +1|\boldsymbol{x}_i) \leq p(y = +1|\boldsymbol{x}_j) \Leftrightarrow r(\boldsymbol{x}_i) \leq r(\boldsymbol{x}_j).$$

A proof is provided in Appendix A.

Even though we cannot estimate $p(y = +1|\cdot)$, Theorem 1 implies that we can still extract the total order in $\mathcal{X}$ induced by $p(y = +1|\cdot)$ if we can estimate $r(\cdot)$. Therefore, we propose to estimate $r$ and use it as a score function that captures the total order induced by $p(y = +1|\cdot)$. After obtaining an estimator of $r$ (denoted by $\hat{r}$), we set a threshold $\theta \in \mathbb{R}$ and use $h(\boldsymbol{x}) = \mathtt{sign}(r(\boldsymbol{x}) - \theta)$ as a classifier. There are various ways of determining the threshold. For instance, we put labels from data with the highest density ratio under a constraint on the number of data to which we can put labels (Hido et al., 2011). Here, we introduce one useful principle for choosing $\theta$. We consider using a threshold $\theta_\pi$ defined by the following equation,

$$\pi = \int \mathbf{1}[r(\boldsymbol{x}) \geq \theta_\pi]p(\boldsymbol{x})d\boldsymbol{x}. \tag{2}$$

The intuition behind the definition of $\theta_\pi$ is that the proportion of the positive data in the test data points should not deviate so much from the class-prior. This intuition becomes clearer in Section 4.3.

In Section 4, we discuss detailed methods for estimating $r$ and setting $\theta$ based on data. Our approach is summarized in the form of a pseudo-code in Algorithm 1. In the rest of this section, we theoretically justify $\theta_\pi$ defined in (2).

**Property of $\theta_\pi$:** Let us consider the case where a classifier is given as $h(\boldsymbol{x}) = \mathtt{sign}(r(\boldsymbol{x}) - \theta)$. Then four population quantities, *true positives* (TP), *true negatives* (TN), *false positives* (FP), and *false negatives* (FN) (Lipton et al., 2014), that depend on $r(\cdot)$ and $\theta$ is written as follows:

$$TP = \int_{\{\boldsymbol{x}:r(\boldsymbol{x})\geq\theta\}} p(y = +1|\boldsymbol{x})p(\boldsymbol{x})d\boldsymbol{x}, \quad FP = \int_{\{\boldsymbol{x}:r(\boldsymbol{x})\geq\theta\}} p(y = -1|\boldsymbol{x})p(\boldsymbol{x})d\boldsymbol{x},$$

$$TN = \int_{\{\boldsymbol{x}:r(\boldsymbol{x})<\theta\}} p(y = -1|\boldsymbol{x})p(\boldsymbol{x})d\boldsymbol{x}, \quad FN = \int_{\{\boldsymbol{x}:r(\boldsymbol{x})<\theta\}} p(y = +1|\boldsymbol{x})p(\boldsymbol{x})d\boldsymbol{x}.$$

Then, the *precision* and the *recall* of a classifier are expressed as precision $= \left(\frac{TP}{TP+FP}\right)$ and recall $= \left(\frac{TP}{TP+FN}\right)$, respectively. For $\theta_\pi$, we have the following result.

**Theorem 2.** *If we use $\theta = \theta_\pi$, then* precision $=$ recall *holds.*

A proof is shown in Appendix B. The threshold $\theta_\pi$ is known as *precision–recall breakeven point* (BEP) (Sammut & Webb, 2010), which makes the precision and the recall the same. BEP is originally used to evaluate a generic classification model with a score function and a threshold. Besides, we can interpret BEP as a point which balances a prediction result; as explained by Powers (2015), a classifier using BEP as a threshold puts the same cost to the false positives and false negatives. Knowing BEP is also useful for deciding on a threshold which put unbalanced weight on the precision and the recall because we can tell if we are weighing precision more or recall more.

## 4 ALGORITHM

Here, we propose two directions for estimating $r(\boldsymbol{x}) = \frac{p(\boldsymbol{x}|y=+1,o=+1)}{p(\boldsymbol{x})}$ under the assumption of invariance of order, namely minimizing a *pseudo classification risk* and *direct density ratio estimation*. We discuss how to set $\theta$ based on the data. A pseudo-code of our algorithm is shown in Algorithm 2.

---

**Algorithm 2** PUSB

---

**Input:** A positive dataset $\{\boldsymbol{x}_i\}_{i=1}^n$, an unlabeled dataset $\{\boldsymbol{x}_i'\}_{i=1}^{n'}$, a test dataset $\{\boldsymbol{x}_i^{\text{te}}\}_{i=1}^{n^{\text{te}}}$ and the class-pror $\pi$.

Using $\{\boldsymbol{x}_i\}_{i=1}^n$ and $\{\boldsymbol{x}_i'\}_{i=1}^{n'}$, estimate $r(\boldsymbol{x})$ by any of (5), (6) or (8) and obtain $\hat{r}(\boldsymbol{x})$.

Using $\hat{r}(\boldsymbol{x})$, estimate $\theta_\pi$ by (9) and obtain $\hat{\theta}_\pi$.

Using an estimator $\hat{r}(\boldsymbol{x})$ and $\hat{\theta}$, obtain a classifier $h(\boldsymbol{x}) = \texttt{sign}(\hat{r}(\boldsymbol{x}) - \hat{\theta}_\pi)$.

---

### 4.1 ESTIMATION OF $r$ BY MINIMIZING PSEUDO CLASSIFICATION RISK

First, we introduce the minimization of the pseudo classification risk. The idea is to minimize the classification risk used in traditional PU learning (du Plessis et al., 2014; 2015) as if there is no selection bias. Under a selection bias, we cannot construct unbiased risk function, but the minimizer can be substituted for the density ratio in (1).

**Conventional PU risk formulation:** Let $\ell : \mathbb{R} \times \{\pm 1\} \to \mathbb{R}^+$ be a loss function, where $\mathbb{R}^+$ is the set of non-negative real values, and $\mathcal{F}$ be the set of measurable functions from $\mathcal{X}$ to $[\epsilon, 1 - \epsilon]$, where $\epsilon \in (0, 1/2)$ is a small positive value. This constant $\epsilon$ is introduced to make the following optimization problem well-defined.

du Plessis et al. (2015) showed that the classification risk of $f \in \mathcal{F}$ in the traditional PU problem setting with SCAR can be expressed as

$$R_{\text{PU}}(f) = \pi \mathbb{E}_{\text{p}}[\ell(f(X), +1)] - \pi \mathbb{E}_{\text{p}}[\ell(f(X), -1)] + \mathbb{E}_{\text{u}}[\ell(f(X), -1)], \tag{3}$$

where $\mathbb{E}_{\text{p}}$ and $\mathbb{E}_{\text{u}}$ are the expectations over $p(\boldsymbol{x}|y = +1)$ and $p(\boldsymbol{x})$, respectively. When there is no selection bias, we can replace the expectations with the corresponding sample averages to obtain an unbiased estimator of the classification risk.

**The pseudo classification risk:** In our problem setting, we only have samples from $p(\boldsymbol{x}|y = +1, o = +1)$ and not from $p(\boldsymbol{x}|y = +1)$. Therefore, we cannot use our sample to obtain an empirical version of $R_{\text{PU}}$. However, we still consider the *pseudo classification risk* of $f \in \mathcal{F}$:

$$R_{\text{PU}}^{\texttt{bias}}(f, \ell) = \pi \mathbb{E}_{\text{p}}^{\texttt{bias}}[\ell(f(X), +1)] - \pi \mathbb{E}_{\text{p}}^{\texttt{bias}}[\ell(f(X), -1)] + \mathbb{E}_{\text{u}}[\ell(f(X), -1)],$$

where $\mathbb{E}_{\text{p}}^{\texttt{bias}}$ is the expectation over $p(\boldsymbol{x}|y = +1, o = +1)$. We call this functional the pseudo classification risk because it is not the true classification risk. An unbiased estimator for the pseudo classification risk can be obtained by replacing the expectations with the corresponding sample averages even if there is a selection bias. For the loss function, we use the logarithmic loss: $\ell(f(\boldsymbol{x}), +1)) = -\log(f(\boldsymbol{x}))$ and $\ell(f(\boldsymbol{x}), -1) = -\log(1 - f(\boldsymbol{x}))$. In this case, the pseudo classification risk of $f \in \mathcal{F}$ becomes

$$R_{\text{PU}}^{\texttt{bias}}(f) = -\pi \mathbb{E}_{\text{p}}^{\texttt{bias}}[\log(f(X))] + \pi \mathbb{E}_{p}^{\texttt{bias}}[\log(1 - f(X))] - \mathbb{E}_{\text{u}}[\log(1 - f(X))]. \tag{4}$$

**Justification for minimizing the pseudo classification risk:** For the pseudo classification risk with the logarithmic loss function, the following theorem justifies its use. Let us denote a minimizer of (4) by $f^*$, that is,

$$f^* \in \arg\min_{f \in \mathcal{F}} R_{\text{PU}}^{\text{bias}}(f).$$

For the minimizer of (4), we show the following theorem.

**Theorem 3.** *It holds almost everywhere that*

$$f^*(\boldsymbol{x}) = \begin{cases} \epsilon & (\boldsymbol{x} \notin D_1), \\ \frac{\pi p(\boldsymbol{x}|y=+1, o=+1)}{p(\boldsymbol{x})} & (\boldsymbol{x} \in D_1 \cap D_2), \\ 1 - \epsilon & (\boldsymbol{x} \notin D_2), \end{cases}$$

*where $D_1 = \{\boldsymbol{x} | \pi p(\boldsymbol{x}|y = 1, o = +1) \geq \epsilon p(\boldsymbol{x})\}$ and $D_2 = \{\boldsymbol{x} | \pi p(\boldsymbol{x}|y = 1, o = +1) \leq (1 - \epsilon)p(\boldsymbol{x})\}$.*

A proof is provided in Appendix C. Theorem 3 implies that the minimization of the empirical version of the pseudo classification risk allows us to estimate $r$.

**Empirical Estimation:** When we train a classifier with training samples, we can naively replace the expectations with the corresponding sample averages. For a hypothesis set $\mathcal{H}$, which is a set of measurable functions, let us define the following risk minimization problem,

$$\hat{f}_1 = \underset{f \in \mathcal{H}}{\arg\min} \left[ -\pi \hat{\mathbb{E}}_p^{\texttt{bias}}[\log(f(X))] + \pi \hat{\mathbb{E}}_p^{\texttt{bias}}[\log(1 - f(X))] - \hat{\mathbb{E}}_u[\log(1 - f(X))] + \mathcal{R}(f) \right],$$

(5)

where $\hat{\mathbb{E}}_p^{\texttt{bias}}$ denotes the averaging operator over positive data with a selection bias, $\hat{\mathbb{E}}_u$ denotes the averaging over the unlabeled data, and $\mathcal{R}$ is a regularization term. du Plessis et al. (2015) showed that, under SCAR, the empirical version of the risk becomes unbiased toward the classification risk.

However, Kiryo et al. (2017) pointed out that unbiased PU learning does not work with deep neural networks. Minimizing an empirical risk of (3) with deep neural networks easily causes over-fitting because the risk is not bounded from below by $0$. In order to implement PU learning with deep neural networks, Kiryo et al. (2017) proposed the following non-negative risk,

$$\hat{f}_2 = \underset{f \in \mathcal{H}}{\arg\min} \left[ -\pi \hat{\mathbb{E}}_p^{\texttt{bias}}[\log(f(X))] + \left( \pi \hat{\mathbb{E}}_p^{\texttt{bias}}[\log(1 - f(X))] - \hat{\mathbb{E}}_u[\log(1 - f(X))] \right)_+ + \mathcal{R}(f) \right],$$

(6)

where $(\cdot)_+ := \max\{0, \cdot\}$. After obtaining $\hat{f}$, we construct an estimator of the density ratio $r$ by $\hat{r} = \frac{1}{\pi}\hat{f}_1$ or $\hat{r} = \frac{1}{\pi}\hat{f}_2$.

## 4.2 ESTIMATION OF $r$ BY DIRECT DENSITY RATIO ESTIMATION

For another approach, we consider estimating the density ratio $r(\boldsymbol{x}) = \frac{p(\boldsymbol{x}|y=+1,o=+1)}{p(\boldsymbol{x})}$ directly. We can estimate the probability density functions of the numerator and the denominator. However, as known as Vapnik's principle, we should avoid solving more difficult intermediate problems than the target problem. Sugiyama et al. (2012) summarized methods estimating the density ratio directly. Among existing methods, we employ *Least-squares importance fitting* (LSIF), which uses the squared loss for density-ratio function fitting. The reason for this choice is that there is an algorithm called *unconstrained Least-Squares Importance Fitting* (uLSIF) with a computational advantage. We can obtain the closed-form solution just by solving the linear equations. Thus, uLSIF is numerically stable when it is regularized properly. Moreover, the leave-one-out cross-validation score for uLSIF can also be computed analytically, which significantly improves the computational efficiency in model selection.

Here, we introduce the formulation of LSIF. Let $\mathcal{S}$ be the class of non-negative measurable functions $s : \mathcal{X} \to \mathbb{R}^+$. We consider minimizing the following squared error between $s$ and $r$:

$$R_{\text{DR}}(s) = \mathbb{E}_u[(s(X) - r(X))^2] = \mathbb{E}_u[(r(X))^2] - 2\mathbb{E}_p^{\texttt{bias}}[s(X)] + \mathbb{E}_u[(s(X))^2]. \tag{7}$$

The first term of the last equation does not affect the result of minimization and we can ignore the term, i.e., the density ratio is estimated through the following minimization problem:

$$s^* = \underset{s \in \mathcal{S}}{\arg\min} \, R_{\text{DR}}(s) = \underset{s \in \mathcal{S}}{\arg\min} \left[ \frac{1}{2}\mathbb{E}_u[(s(X))^2] - \mathbb{E}_p^{\texttt{bias}}[s(X)] \right].$$

**Empirical Estimation:** As mentioned above, to minimize the empirical version of (7), we use uLSIF (Kanamori et al., 2009). Given a hypothesis class $\mathcal{H}$, we obtain $\hat{r}$ by

$$\hat{r} = \underset{s \in \mathcal{H}}{\arg\min} \left[ \frac{1}{2}\hat{\mathbb{E}}_u[(s(X))^2] - \hat{\mathbb{E}}_p^{\texttt{bias}}[s(X)] + \mathcal{R}(s) \right], \tag{8}$$

where $\mathcal{R}$ is a regularization term.

## 4.3 ESTIMATION OF $\theta_\pi$

We consider replacing the threshold defined by (2) with a value which can be calculated only from samples. By using the test inputs or held-out training data, $\{\boldsymbol{x}_i^{\text{te}}\}_{i=1}^{n^{\text{te}}} \sim p(\boldsymbol{x})$, we find $\hat{\theta}_\pi$ that satisfies

Table 1: Dataset statistics (Pos. frac.: Positive fraction, Dim: Dimension).

| Dataset | # of samples | Pos. frac. | Dim. |
|---|---|---|---|
| mushrooms | 8,124 | 0.517 | 112 |
| shuttle | 58,000 | 0.786 | 9 |
| pageblocks | 5,473 | 0.898 | 10 |
| usps | 9,298 | 0.524 | 256 |
| connect-4 | 67,557 | 0.658 | 126 |
| spambase | 4,601 | 0.394 | 57 |
| MNIST | 70,000 | 0.511 | 784 |
| CIFAR-10 | 60,000 | 0.400 | 3,072 |

the following equation,

$$\lceil \pi n^{\text{te}} \rceil = \sum_{i=1}^{n^{\text{te}}} \mathbf{1}[\hat{r}(\boldsymbol{x}_i^{\text{te}}) > \hat{\theta}_\pi].\tag{9}$$

Here, we used the knowledge of $\pi$, the class-prior. This choice of $\hat{\theta}_\pi$ amounts to classifying top-$\pi$ test data as positive after ranking the inputs by $\hat{r}$.

## 5 EXPERIMENTS

In this section, we report experimental results which were conducted using synthetic data and real-world datasets[1]. We used seven classification datasets, mushrooms, shuttle, pageblocks, usps, connect-4, spambase, and MNIST, from UCI repository[2], CIFAR-10[3] and a document dataset obtained from SwissProt (Boeckmann et al., 2003)[4]. MNIST and CIFAR-10 originally have 10 and 10 classes, respectively, and we constructed the positive and negative datasets from them as follows: MNIST was preprocessed in such a way that 0, 2, 4, 6, 8 constitute the positive class, while 1, 3, 5, 7, 9 constitute the negative class; for CIFAR-10, the positive dataset is formed by 'airplane', 'automobile', 'ship' and 'truck', and the negative dataset is formed by 'bird', 'cat', 'deer', 'dog', 'frog' and 'horse'. Except for the document dataset, we show the details of datasets in Table 1 and made positive data with a selection bias based on estimators of $p(y = +1|\boldsymbol{x})$ as we show in each experiments. For six datasets of the UCI repository and the CIFAR-10, we made positive datasets with a selection bias artificially, but, for the document dataset, we have an unlabeled dataset and a positive dataset, which is gathered for classifying the labels in the unlabeled dataset.

We call unbiased PU learning proposed by du Plessis et al. (2015) "PU", unbiased PU learning with a threshold estimated by (9) "PUSB", uLSIF with a threshold estimated by (9) "DRSB", nonnegative PU learning "nnPU" and nonnegative PU learning with a threshold estimated by (9) "nnPUSB".

For the hypothesis class $\mathcal{H}$ in the density ratio estimation (8), we use the linear-in-parameter model:
$$\mathcal{H} := \left\{ s(\boldsymbol{x}) = \boldsymbol{\beta}^\top \boldsymbol{\varphi}(\boldsymbol{x}) \,\middle|\, \boldsymbol{\beta} \in \mathbb{R}^{m+1} \right\},\tag{10}$$
where $\boldsymbol{\varphi}(\boldsymbol{x}) = [1, \varphi_1(\boldsymbol{x}), ..., \varphi_m(\boldsymbol{x})]^\top$ is a vector of basis functions. For basis functions, we used the Gaussian kernel located at sample points $\varphi_\ell(\boldsymbol{x}) = \exp\left(-\|\boldsymbol{x} - \boldsymbol{c}_\ell\|^2/(2\sigma^2)\right)$, where $\{\boldsymbol{c}_1, ..., \boldsymbol{c}_m\} = \{\boldsymbol{x}_1, ..., \boldsymbol{x}_n, \boldsymbol{x}'_1, ..., \boldsymbol{x}'_{n'}\}$ and $m = n + n'$. For the hypothesis class $\mathcal{H}$ in the risk minimization of PU learning (5), we use the following model with the sigmoid function:
$$\mathcal{H} := \left\{ f(\boldsymbol{x}) = \frac{1}{1 + \exp(-\boldsymbol{\beta}^\top \boldsymbol{\varphi}(\boldsymbol{x}))} \,\middle|\, \boldsymbol{\beta} \in \mathbb{R}^{m+1} \right\}.$$

In this case, the loss is the same as the logistic loss and unbiased PU learning becomes convex. For the hypothesis class $\mathcal{H}$ in the risk minimization of nonnegative PU learning (6), we use deep neural networks. The specifications of deep neural networks are given in the following sections for each dataset. We mainly used the same structure proposed in Kiryo et al. (2017) in order to compare the performances. For the regularization term $\mathcal{R}$, we used the $\ell_2$ norm of the parameters scaled by a positive scalar $\lambda$. For the linear models, hyperparameters were selected via cross-validation.

---

[1] The source code is available at https://github.com/MasaKat0/PUlearning.

[2] The UCI data were downloaded from https://archive.ics.uci.edu/ml/index.php and https://www.csie.ntu.edu.tw/~cjlin/libsvmtools/.

[3] See https://www.cs.toronto.edu/~kriz/cifar.html.

[4] The data can be downloaded from http://www.cs.ucsd.edu/users/elkan/posonly.

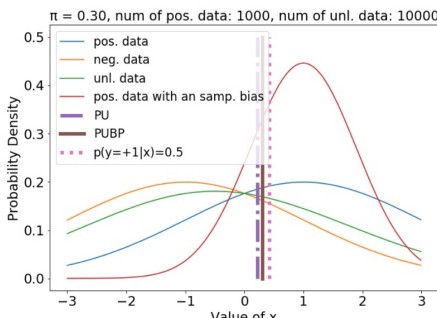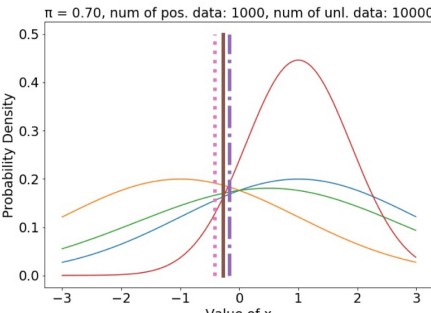

Figure 1: Two Gaussians: The horizontal axis is the value of $\boldsymbol{x}$ and the vertical axis is the probability density. The vertical lines represent the decision boundaries of the classifiers. The distribution of positive data, negative data, unlabeled data and $p(\boldsymbol{x}|o = +1, y = +1)$ are plotted.

## 5.1 TEST WITH SYNTHETIC DATA

This experiment shows the classifier given by PUSB. We used samples from a mixture distribution of the following two class-conditional distributions:

$$p(\boldsymbol{x}|y = +1) = \mathcal{N}(1, 2^2) \text{ and } p(\boldsymbol{x}|y = -1) = \mathcal{N}(-1, 2^2),$$

where $\mathcal{N}(\mu, \sigma^2)$ denotes the univariate normal distribution with mean $\mu$ and variance $\sigma^2$. A positive dataset with a selection bias was sampled from

$$p(\boldsymbol{x}|o = +1, y = +1) \propto (p(y = +1|\boldsymbol{x}))^{10}.$$

We generated $1,000$ positive samples and $10,000$ unlabeled samples. We made two datasets with different class-priors $\pi = 0.3$ and $\pi = 0.7$. Figure 1 shows classifiers constructed by PU and PUSB along with the Bayes optimal classifier $\mathtt{sign}(p(y = +1|\boldsymbol{x}) - 1/2)$. The classifier of PUSB is closer to the Bayes optimal classifier than that of PU.

## 5.2 TEST WITH BENCHMARK DATA

Here, we investigate the experimental performance in detail.

**Linear-in-parameter model:** We used the `mushrooms`, `shuttle`, `pageblocks`, `usps`, `connect-4` and `spambase` datasets. First, we estimated $p(y = +1|\boldsymbol{x})$ using the logistic regression with the same linear model. Then, we obtained the labeled positive data by labeling some instances of the positive data following

$$p(o = +1|\boldsymbol{x}, y = +1) \propto (p(y = +1|\boldsymbol{x}))^{20}.$$

Then, we trained a classifier by minimizing the empirical risk of PU learning (4) and the density ratio estimation (7).

For each binary labeled dataset, we made 12 different pairs of positive and unlabeled data with 4 different class-priors, $\{0.2, 0.4, 0.6, 0.8\}$, and 3 different numbers of unlabeled data, $\{800, 1600, 3200\}$. The number of positive data was fixed at $400$. We used 1000 test data sampled from the same distribution as the unlabeled data. We ran the experiments 100 times and calculated the mean and standard deviation for the test dataset with PU, PUSB, and DRSB. The results are shown in Table 2. The classifiers obtained by our algorithm always show preferable performance to existing methods.

**Neural network model:** We used the `MNIST` and `CIFAR-10` datasets. For `MNIST`, a 3-layer multilayer perceptron (MLP) with ReLU activation (Nair & Hinton, 2010) was used. For `CIFAR-10`, an all convolutional net (Springenberg et al., 2015) was used. Details of the network structure are shown in Appendix D.

Table 2: The error rate of classification in test data (%) are shown for the different class-priors and the different number of samples. For all experiments, the linear-in-parameter model was used. Best and equivalent methods (under 5% t-test) are bold.

| Dataset | $\pi$ | PU | | | PUSB | | | DRSB | | |
|---|---|---|---|---|---|---|---|---|---|---|
| | | 800 | 1600 | 3200 | 800 | 1600 | 3200 | 800 | 1600 | 3,200 |
| mushrooms | 0.2 | 5.1 (.076) | 5.0 (.055) | 5.5 (.083) | **4.1** (.007) | **3.8** (.007) | **4.0** (.007) | 10.7 (.010) | 9.9 (.011) | 9.9 (.012) |
| | 0.4 | **7.0** (.012) | **6.6** (.010) | **6.8** (.025) | **6.9** (.012) | **6.7** (.016) | **6.8** (.010) | 15.8 (.014) | 15.0 (.016) | 15.2 (.012) |
| | 0.6 | 10.8 (.020) | 11.3 (.015) | 11.3 (.016) | **8.2** (.013) | **8.5** (.010) | **8.3** (.011) | 18.5 (.014) | 18.8 (.012) | 18.5 (.013) |
| | 0.8 | 21.1 (.022) | 21.6 (.021) | 21.7 (.016) | **8.1** (.015) | **8.0** (.012) | **7.6** (.014) | 14.2 (.013) | 14.3 (.014) | 14.4 (.011) |
| shuttle | 0.2 | 23.8 (.006) | 23.8 (.007) | 23.8 (.007) | 5.2 (.008) | **5.0** (.007) | **5.0** (.007) | **5.1** (.008) | **5.1** (.008) | 5.1 (.007) |
| | 0.4 | 15.5 (.023) | 15.5 (.019) | 15.0 (.016) | 7.8 (.011) | **7.7** (.011) | 7.6 (.012) | **6.7** (.025) | 7.6 (.029) | 7.2 (.027) |
| | 0.6 | 26.1 (.018) | 26.4 (.017) | 26.4 (.016) | 11.1 (.015) | 10.8 (.012) | 11.0 (.015) | 7.8 (.028) | **7.4** (.028) | **7.1** (.028) |
| | 0.8 | 14.6 (.006) | 14.6 (.005) | 14.8 (.007) | 12.3 (.014) | 12.4 (.014) | 12.3 (.012) | **7.9** (.021) | **7.6** (.023) | **7.4** (.022) |
| pageblocks | 0.2 | 39.5 (.009) | 39.6 (.010) | 39.6 (.008) | 41.6 (.021) | 42.5 (.022) | 43.7 (.019) | **23.1** (.016) | **23.1** (.015) | **22.2** (.012) |
| | 0.4 | 56.5 (.011) | 56.7 (.011) | 56.6 (.008) | 33.7 (.020) | 33.7 (.019) | 33.5 (.024) | **23.6** (.012) | **23.7** (.014) | **23.6** (.014) |
| | 0.6 | 22.4 (.023) | 23.4 (.033) | 28.1 (.028) | 20.9 (.016) | 21.1 (.016) | 21.5 (.020) | **18.7** (.011) | **18.6** (.012) | **18.1** (.017) |
| | 0.8 | 19.8 (.003) | 20.0 (.001) | 20.0 (.001) | **13.9** (.022) | **13.6** (.023) | 16.7 (.043) | 15.4 (.011) | 14.8 (.014) | **14.3** (.019) |
| usps | 0.2 | 9.0 (.011) | 8.5 (.009) | 8.2 (.010) | **8.0** (.009) | **7.7** (.007) | **7.4** (.009) | 18.2 (.016) | 19.6 (.013) | 19.7 (.014) |
| | 0.4 | **10.5** (.012) | **10.3** (.013) | 10.0 (.010) | **10.5** (.013) | **10.2** (.012) | **10.0** (.010) | 30.5 (.029) | 30.2 (.022) | 29.9 (.023) |
| | 0.6 | 12.5 (.015) | 12.3 (.015) | 12.1 (.013) | **11.2** (.016) | **10.9** (.015) | **10.6** (.013) | 32.9 (.026) | 33.2 (.031) | 33.1 (.030) |
| | 0.8 | 19.8 (.020) | 19.8 (.016) | 19.5 (.017) | **10.3** (.016) | **9.8** (.014) | **9.6** (.014) | 25.9 (.028) | 25.3 (.027) | 25.8 (.030) |
| connect-4 | 0.2 | 22.3 (0.017) | **21.9** (.014) | 21.6 (.014) | **21.8** (.013) | **21.5** (.012) | **21.2** (.011) | 26.7 (.012) | 26.6 (.012) | 26.4 (.011) |
| | 0.4 | **31.2** (.015) | **31.0** (.015) | **30.7** (.016) | **31.2** (.015) | **31.0** (.015) | **30.7** (.016) | 39.4 (.016) | 39.6 (.018) | 39.2 (.016) |
| | 0.6 | 32.0 (.017) | **32.1** (.013) | 31.9 (.015 | **31.7** (.016) | **31.7** (.013) | **31.4** (.016) | 40.9 (.015) | 41.1 (.017) | 41.0 (.017) |
| | 0.8 | 33.6 (.018) | 33.5 (.016) | 33.4 (.017) | **24.2** (.013) | **23.9** (.012) | **23.8** (.013) | 29.8 (.011) | 29.6 (.013) | 29.5 (.012) |
| spambase | 0.2 | 20.1 (0.002) | 20.1 (.002) | 20.1 (.003) | **13.6** (.013) | **13.8** (.014) | **14.0** (.014) | 18.1 (.026) | 18.3 (.025) | 17.9 (.023) |
| | 0.4 | 36.2 (.024) | 35.9 (.025) | 30.7 (.024) | **19.0** (.020) | **18.7** (.021) | **18.9** (.018) | 27.4 (.042) | 27.8 (.043) | 27.7 (.039) |
| | 0.6 | 40.0 (.001) | 39.9 (.001) | 31.9 (.001 | **20.8** (.019) | **20.7** (.018) | **20.0** (.017) | 31.2 (.037) | 30.3 (.035) | 31.2 (.035) |
| | 0.8 | 20.0 (.000) | 20.0 (.000) | 20.0 (.000) | **18.0** (.015) | **17.7** (.013) | **17.3** (.013) | 24.5 (.058) | 23.9 (.017) | 24.2 (.017) |

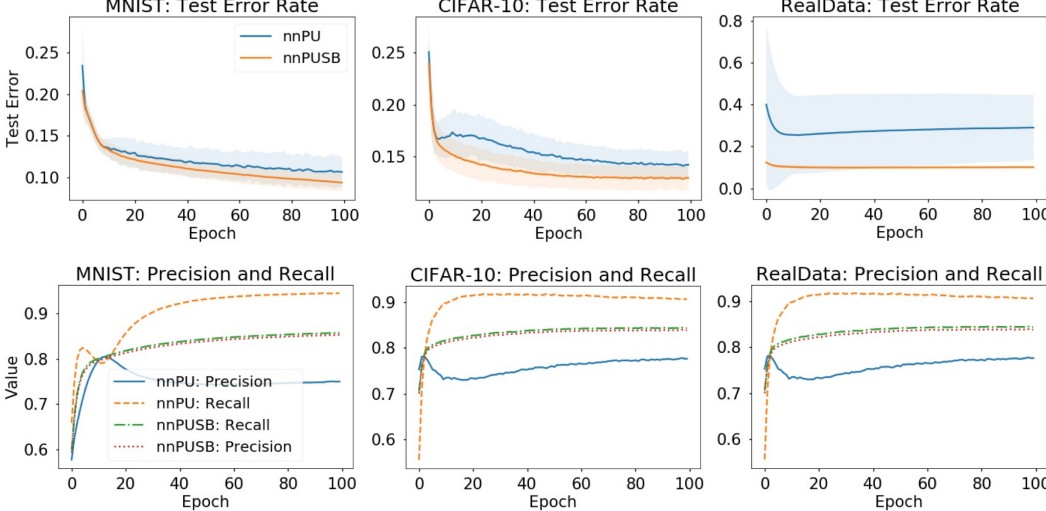

Figure 2: Experimental results of training deep neural networks. Left: MNIST; Center: CIFAR-10; Right: RealData: All measures are calculated for test data sampled from the marginal distribution $p(\boldsymbol{x})$. The horizontal axis is the epoch of training the network, the vertical axes of the top figures are the error rates and the vertical axes of the bottom figures are the precision and the recall.

First, we estimated $p(y = +1|\boldsymbol{x})$ using the logistic regression with the same network structure using the positive and negative datasets in the unlabeled dataset. Next, from the positive dataset, we resampled positive dataset with an observation, which follows

$$p(o = +1|\boldsymbol{x}, y = +1) \propto (p(y = +1|\boldsymbol{x}))^{10}.$$

Then, we trained a classifier by minimizing (6) with the model defined above. We used $10,000$ test data sampled from the same distribution as the unlabeled data. We ran the experiments 100 times and calculated the mean of the error rate, the standard deviation of the error rate, the mean of recall and the mean of precision for each epoch in training with nnPU and nnPUSB. The results are shown on the left side and center of Figure 2. In the upper row, we show the mean and standard deviation of the error rate. In the lower row, we show the mean of recall and the mean of precision. As shown in Figure 2, the mean of the error rate of our algorithm is lower and the variance is also lower than the existing method. As we discussed in Section 4.3, $\mathrm{FP} = \mathrm{FN}$ is also empirically observed.

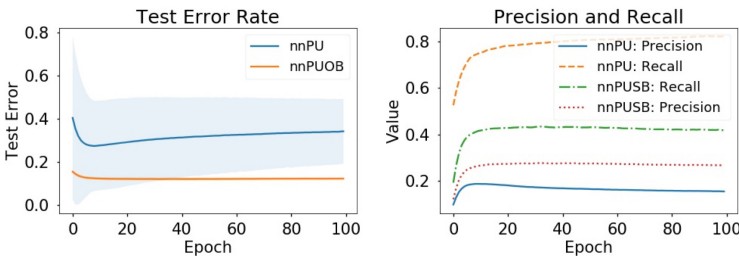

Figure 3: Results of the second experiment of Section 5.4 using the `RealData` dataset with the estimated class prior $0.1095$ (the true class prior is $0.0709$). The horizontal axes are the epochs of training the network, the vertical axis of the right figure is the error rate and the vertical axis of the left figure is the value of the precision and the recall.

## 5.3 TEST WITH REAL-WORLD DATA

In the previous sections, we artificially made positive data with a selection bias. Here we demonstrate the effectiveness of our algorithm in real-world data with a selection bias. We used a document dataset based on the SwissProt database released by Elkan & Noto (2008). We call this dataset `RealData`. The dataset originally contained 2,453 labeled positive examples (P) and 4,906 unlabeled examples (U). The unlabeled examples were labeled later by Das et al. (2007). As a result, the dataset is likely to have a natural observation bias in the P data while it allows an access to the ground-truth labels for all data. Out of the U data, 348 examples are positive and the rest are negative. The class prior is $\pi = 348/4,906 = 0.0709$. We used Bag-of-Words to represent the documents as $78,894$-dimensional vectors.

Details of the network structure are shown in Appendix D. We trained a classifier using positive and unlabeled data. Then, after finding a threshold estimated by (9), we evaluated the same evaluation measures as the previous experiments by classifying U, i.e., we regarded the unlabeled data as test data. As shown on the right of Figure 2, the result of our algorithm outperforms the existing method.

## 5.4 TEST FOR UNKNOWN CLASS PRIOR

In order to evaluate how our algorithm works for the case that the class prior is unknown, we empirically tested our algorithm with an estimator of the class prior. We used the `RealData` dataset in Section 5.3, whose class prior is $0.0709$. For the class prior estimator, we used the KM2 method by Ramaswamy et al. (2016), which is considered to be the state-of-the-art method in the case-control scenario under SCAR. The estimated class prior was $0.1095$ and the result is shown in Figure 3. We can see from the figure that our method works well with an estimated class prior in so far as `RealData` dataset is concerned. We also show results for the classifiers trained under misspecified class priors in Appendix E, which also shows that our method works stably for misspecified class priors.

## 6 CONCLUSION

In this paper, we proposed a novel framework for PU learning with a selection bias in positive data. We put the assumption of the invariance of order and showed the density ratio of labeled positive data and unlabeled data has the same order as the class-conditional distribution for inputs. Based on this result, we proposed a method based on partial identification in which we first estimate the density ratio and then use it as a classifier by setting a threshold. We conducted experiments to confirm the effectiveness of our approach. As we showed in the experiments, our method outperforms previous PU methods on real-world data.

ACKNOWLEDGMENTS

This work was supported by JSPS KAKENHI 16H00881 and the AIP challenge program, Japan.

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

## A    PROOF OF THEOREM 1

*Proof.* We assumed that no negative data can be labeled. As a result, we have $p(o = +1|\boldsymbol{x}, y = -1) = 0$ for arbitrary $x \in \mathcal{X}$. Therefore,

$$
\begin{aligned}
p(o = +1|\boldsymbol{x}) &= p(y = +1|\boldsymbol{x})p(o = +1|\boldsymbol{x}, y = +1) + p(y = -1|\boldsymbol{x})p(o = +1|\boldsymbol{x}, y = -1) \\
&= p(y = +1|\boldsymbol{x})p(o = +1|\boldsymbol{x}, y = +1).
\end{aligned}
\tag{11}
$$

By Bayes' theorem, we can expand the density ratio in (1) as follows:

$$
\begin{aligned}
r(\boldsymbol{x}) &= \frac{p(\boldsymbol{x}|o = +1, y = +1)}{p(\boldsymbol{x})} \\
&= \frac{p(y = +1|\boldsymbol{x}, o = +1)p(\boldsymbol{x}|o = +1)}{p(y = +1|o = +1)} \frac{1}{p(\boldsymbol{x})} \\
&= p(y = +1|\boldsymbol{x}, o = +1)\frac{1}{p(y = +1|o = +1)} \frac{p(\boldsymbol{x}|o = +1)}{p(\boldsymbol{x})} \\
&= \frac{p(y = +1|\boldsymbol{x})p(o = +1|\boldsymbol{x}, y = +1)}{p(o = +1|\boldsymbol{x})} \frac{1}{p(y = +1|o = +1)p(o = +1)} \frac{p(o = +1)p(\boldsymbol{x}|o = +1)}{p(\boldsymbol{x})}.
\end{aligned}
$$

Because $\frac{p(y=+1|\boldsymbol{x})p(o=+1|\boldsymbol{x},y=+1)}{p(o=+1|\boldsymbol{x})} = 1$ from (11), this is equivalent to

$$
r(\boldsymbol{x}) = Cp(o = +1|\boldsymbol{x}),
$$

where $C = \frac{1}{p(y=+1,o=+1)}$. Hence, if Assumption 1 holds, for any $\boldsymbol{x}_i, \boldsymbol{x}_j \in \mathcal{X}$,

$$
p(y = +1|\boldsymbol{x}_i) \leq p(y = +1|\boldsymbol{x}_j) \Leftrightarrow r(\boldsymbol{x}_i) \leq r(\boldsymbol{x}_j).
$$

$\square$

## B PROOF OF THEOREM 2

Before proving Theorem 2, we consider a threshold defined as follows:

$$
\pi = \int \mathbf{1}[p(y = +1|\boldsymbol{x}) \geq \gamma]p(\boldsymbol{x})d\boldsymbol{x}. \tag{12}
$$

Then, we state the following lemma on the relationship between $\gamma$ and $\theta_\pi$.

**Lemma 1.** *The equation $\boldsymbol{x} \in \mathcal{X}$, $(r(\boldsymbol{x}) - \theta_\pi)(p(y = +1|\boldsymbol{x}) - \gamma) \geq 0$ holds almost everywhere with respect to $p(\boldsymbol{x})$.*

*Proof.* By (2) and (12), the following equation also hold,

$$
\int \mathbf{1}[r(\boldsymbol{x}) > \theta_\pi]p(\boldsymbol{x})d\boldsymbol{x} = \int \mathbf{1}[p(y = +1|\boldsymbol{x}) > \gamma]p(\boldsymbol{x})d\boldsymbol{x}.
$$

Hence, we can derive

$$
\int (\mathbf{1}[r(\boldsymbol{x}) > \theta_\pi] - \mathbf{1}[p(y = +1|\boldsymbol{x}) > \gamma])p(\boldsymbol{x})d\boldsymbol{x} = 0.
$$

This is equivalent to

$$
\int (\mathbf{1}[(r(\boldsymbol{x}) - \theta_\pi)(p(y = +1|\boldsymbol{x}) - \gamma) < 0]p(\boldsymbol{x})d\boldsymbol{x} = 0.
$$

From this equation, $(r(\boldsymbol{x}) - \theta_\pi)(p(y = +1|\boldsymbol{x}) - \gamma) \geq 0$ holds almost surely. $\square$

Using Lemma 1 we prove Theorem 2.

*Proof.* $\pi = \int \mathbf{1}[p(y = +1|\boldsymbol{x}) \geq \gamma]p(\boldsymbol{x})d\boldsymbol{x}$ is equivalent to

$$
\int p(y = +1|\boldsymbol{x})p(\boldsymbol{x})d\boldsymbol{x} = \int \mathbf{1}[p(y = +1|\boldsymbol{x}) \geq \gamma]p(\boldsymbol{x})d\boldsymbol{x},
$$

where the left hand side is equal to

$$
\int_{\{\boldsymbol{x}|p(y=+1|\boldsymbol{x})<\gamma\}} p(y = +1|\boldsymbol{x})p(\boldsymbol{x})d\boldsymbol{x} + \int_{\{\boldsymbol{x}|p(y=+1|\boldsymbol{x})\geq\gamma\}} p(y = +1|\boldsymbol{x})p(\boldsymbol{x})d\boldsymbol{x},
$$

and the right hand side is equal to

$$\int_{\{\boldsymbol{x}|p(y=+1|\boldsymbol{x})\geq\gamma\}} p(y=+1|\boldsymbol{x})p(\boldsymbol{x})d\boldsymbol{x} + \int_{\{\boldsymbol{x}|p(y=+1|\boldsymbol{x})\geq\gamma\}} p(y=-1|\boldsymbol{x})p(\boldsymbol{x})d\boldsymbol{x}.$$

Hence, (12) is equivalent to the following equation,

$$\int_{\{\boldsymbol{x}|p(y=+1|\boldsymbol{x})<\gamma\}} p(y=+1|\boldsymbol{x})p(\boldsymbol{x})d\boldsymbol{x} = \int_{\{\boldsymbol{x}|p(y=+1|\boldsymbol{x})\geq\gamma\}} p(y=-1|\boldsymbol{x})p(\boldsymbol{x})d\boldsymbol{x}.$$

From the definitions of TP, FP, TN, and FN, the left hand side of the above equation is equal to $FP$ and the right hand side of the above equation is equal to $FN$. Hence, $FP = FN$. We showed $FP = FN$ for a threshold $\gamma$, but we can also insist that the same result holds when we use $r(\boldsymbol{x})$ as a score function and $\theta_\pi$ as a threshold. According to Lemma 1, the sign of $p(y = +1 + \boldsymbol{x}) - \gamma$ and $r(\boldsymbol{x}) - \theta_\pi$ are the same almost everywhere with respect to $p(\boldsymbol{x})$. Therefore,

$$\int_{\{\boldsymbol{x}|p(y=+1|\boldsymbol{x})<\gamma\}} p(y=+1|\boldsymbol{x})p(\boldsymbol{x})d\boldsymbol{x} = \int_{\{\boldsymbol{x}|r(\boldsymbol{x})<\theta_\pi\}} p(y=+1|\boldsymbol{x})p(\boldsymbol{x})d\boldsymbol{x}$$

$$\int_{\{\boldsymbol{x}|p(y=+1|\boldsymbol{x})\geq\gamma\}} p(y=-1|\boldsymbol{x})p(\boldsymbol{x})d\boldsymbol{x} = \int_{\{\boldsymbol{x}|r(\boldsymbol{x})\geq\theta_\pi\}} p(y=-1|\boldsymbol{x})p(\boldsymbol{x})d\boldsymbol{x}.$$

In these equations, the right hand sides mean $FP$ and $FN$ of a score function $r(\boldsymbol{x})$ with a threshold $\theta_\pi$, respectively. Because precision $= \left(\frac{TP}{TP+FP}\right)$ and recall $= \left(\frac{TP}{TP+FN}\right)$, $FP = FN$ means that precision = recall. $\qquad\square$

## C  PROOF OF THEOREM 3

*Proof.* We first consider minimizing $R_{\text{PU}}^{\text{bias}}$ in the space of all functions from $\mathcal{X}$ to $[\epsilon, 1 - \epsilon]$ instead of minimizing it in $\mathcal{F}$. Later, we will see that the minimizer matches $f^*$ as stated in Theorem 3 which belongs to $\mathcal{F}$. The minimization of

$$R_{\text{PU}}^{\text{bias}}(f) = \int \left(-\pi p(\boldsymbol{x}|y = +1, o = +1)(\log f(\boldsymbol{x}) - \log(1 - f(\boldsymbol{x}))) - \log(1 - f(\boldsymbol{x}))p(\boldsymbol{x})\right) d\boldsymbol{x}$$

over all functions $f$ taking values in $[\epsilon, 1 - \epsilon]$ is reduced to the following point-wise minimization problem

$$\underset{z\in[\epsilon,1-\epsilon]}{\arg\min} C(z, \boldsymbol{x}) := -\pi p(\boldsymbol{x}|y = +1, o = +1)(\log z - \log(1 - z)) - \log(1 - z)p(\boldsymbol{x}).$$

Denoting the solution by $z^*$, the Karush-Kuhn-Tucker (KKT) condition of this minimization problem is

$$\pi p(\boldsymbol{x}|y = +1, o = +1)\left(\frac{1}{z^*} + \frac{1}{1 - z^*}\right) - \frac{p(\boldsymbol{x})}{1 - z^*} - \lambda + \mu = 0,$$

$$\lambda(z^* - 1 - \epsilon) = 0, \mu z^* = 0,$$

$$\lambda, \mu \geq 0,$$

$$z^* \in [\epsilon, 1 - \epsilon],$$

where $\lambda$ and $\mu$ are the Lagrange multipliers. The first equation is equivalent to

$$(\mu - \lambda)(z^*)^2 + (p(\boldsymbol{x}) + \lambda - \mu)z^* - \pi p(\boldsymbol{x}|y = +1, o = +1) = 0. \qquad (13)$$

We investigate the solution by dividing the KKT condition into the following four cases.

1. If we assume $\lambda = \mu = 0$, then the KKT condition is reduced to

$$z^* = \frac{\pi p(\boldsymbol{x}|y = +1, o = +1)}{p(\boldsymbol{x})},$$

$$z^* \in [\epsilon, 1 - \epsilon].$$

In this case, $z^* \in [\epsilon, 1 - \epsilon]$ is equivalent to $\boldsymbol{x} \in D_1 \cap D_2$.

2. If we assume $\lambda > 0$ and $\mu = 0$, the KKT condition is reduced to

$$z^* = 1 - \epsilon,$$
$$\lambda = \frac{(1-\epsilon)p(\boldsymbol{x}) - \pi p(\boldsymbol{x}|y = +1, o = +1)}{(1-\epsilon)^2 + (1-\epsilon)}.$$

In this case, $\lambda > 0$ is equivalent to $\boldsymbol{x} \notin D_2$.

3. If we assume $\lambda = 0$ and $\mu > 0$, the KKT condition is reduced to

$$z^* = \epsilon,$$
$$\mu = \frac{p(\boldsymbol{x})\epsilon - \pi p(\boldsymbol{x}|y = +1, o = +1)}{\epsilon^2 - \epsilon}.$$

In this case, $\lambda > 0$ is equivalent to $\boldsymbol{x} \notin D_1$.

4. If $\lambda, \mu > 0$, there is no feasible solution.

In summary, the solution for the optimization problem $\arg\min_{z \in [\epsilon, 1-\epsilon]} C(z, \boldsymbol{x})$ is

$$z^* = \begin{cases} \epsilon & (\boldsymbol{x} \notin D_1), \\ \frac{\pi p(\boldsymbol{x}|y=+1, o=+1)}{p(\boldsymbol{x})} & (\boldsymbol{x} \in D_1 \cap D_2), \\ 1 - \epsilon & (\boldsymbol{x} \notin D_2). \end{cases}$$

Finally, we define $f^*(\boldsymbol{x}) := \arg\min_{z \in [\epsilon, 1-\epsilon]} C(z, \boldsymbol{x})$. It can be confirmed that $f^* \in \mathcal{F}$ because $p(\boldsymbol{x}|y = 1, o = +1)$ and $p(\boldsymbol{x})$ are measurable and $f^*$ takes values in $[\epsilon, 1-\epsilon]$. Therefore, the solution of the original optimization problem

$$\arg\min_{f \in \mathcal{F}} R_{\mathrm{PU}}^{\mathtt{bias}}(f)$$

is equal to $f^*$ almost everywhere. $\qquad\square$

## D   NETWORK STRUCTURE USED IN SECTIONS 5.2 AND SECTIONS 5.3

In Section 5.2, we used the `MNIST` and `CIFAR-10` datasets. The model for the `MNIST` dataset was a 3-layer *multilayer perceptron* (MLP) with ReLU (Nair & Hinton, 2010) (more specifically, 784-100-1). The model for the `CIFAR-10` dataset was an all convolutional net (Springenberg et al., 2015): $(32 \times 32 \times 3)$-$[C(3 \times 3, 96)] \times 2$-$C(3 \times 3, 96, 2)$-$[C(3 \times 3, 192)] \times 2$-$C(3 \times 3, 192, 2)$-$C(3 \times 3, 192)$-$C(1 \times 1, 192)$-$C(1 \times 1, 10)$-1000-1000-1, where the input is a $32 \times 32$ RGB image, $C(3 \times 3, 96)$ means 96 channels of $3 \times 3$ convolutions followed by ReLU, $[\cdot] \times 2$ means there are two such layers, $C(3 \times 3, 96, 2)$ means a similar layer but with stride 2, etc.; it is one of the best architectures for `CIFAR-10`. Batch normalization (Ioffe & Szegedy, 2015) was applied before hidden layers.

In Section 5.3, the model for this dataset was a 5-layer *multilayer perceptron* (MLP) with ReLU (more specifically, 78894-300-300-300-300-1).

## E   EXPERIMENTAL RESULTS OF THE FIRST EXPERIMENT OF SECTION 5.4

In the first experiment, we trained a classifier under misspecified class priors, $0.0209(= 0.0709 - 0.0500)$, $0.1209(= 0.0709 + 0.0500)$, $0.1709(= 0.0709 + 0.1000)$, and $0.2709(= 0.0709 + 0.2000)$. The results are shown in Figures 4–6. In Figure 4, the test error in each misspecified class priors are shown. In Figure 5, the precision and recall in each misspecified class prior is shown. In both the existing method and our method, misspecified class priors had a bad influence. However, our method was less influenced by the misspecification and showed better performance than the existing method. Besides, the difference between the precision and recall of our algorithm is narrower than that of the existing method as expected. In order to discuss how misspecified class prior affects the precision and recall, we show the difference the recall from the precision and show the values in Figure 6. This result shows how the difference broadens as the class prior is misspecified worse.

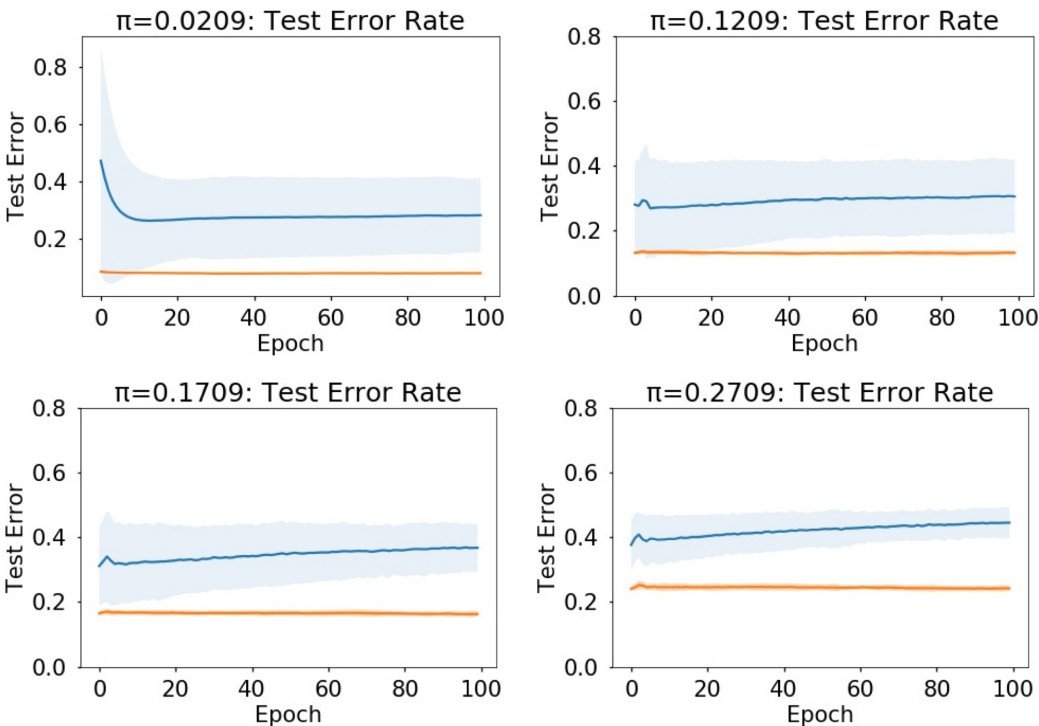

Figure 4: Experimental results of training deep neural networks using the `RealData` dataset with misspecified class priors (the true class prior is $0.0709$). Upper Left: $\pi = 0.0209$; Upper Right: $\pi = 0.1209$; Lower Left: $\pi = 0.1709$; Lower Right: $\pi = 0.2709$: All measures are calculated for test data sampled from the marginal distribution $p(\boldsymbol{x})$. The horizontal line is epoch of training the network, the vertical line of the upper is error rate.

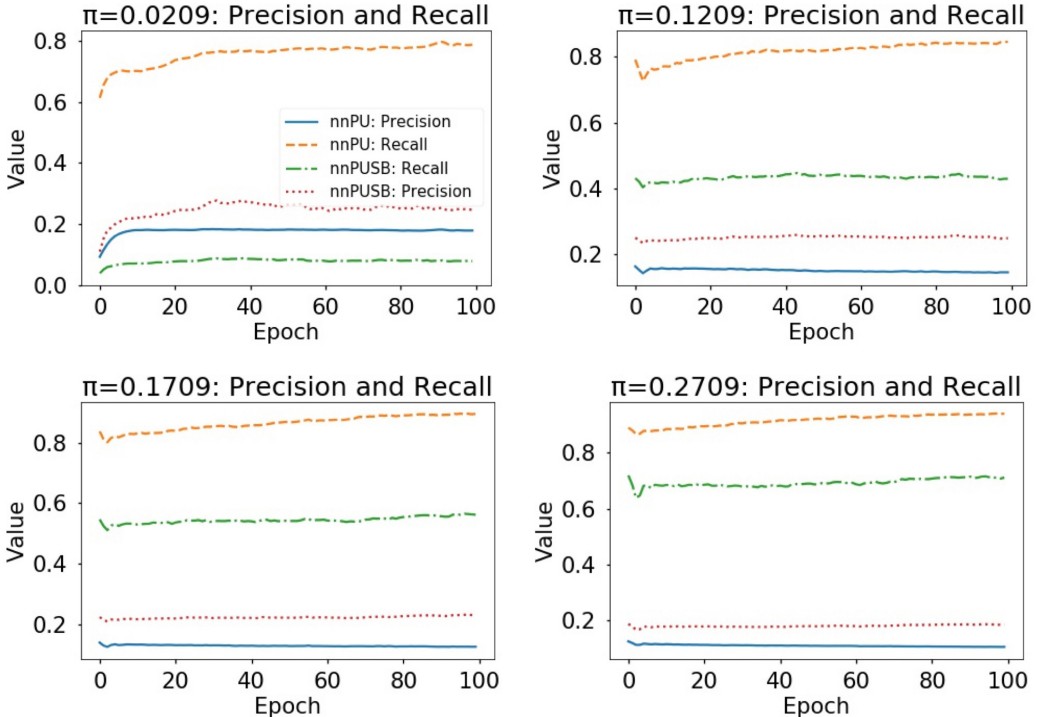

Figure 5: Experimental results of training deep neural networks using the `RealData` dataset with misspecified class priors (the true class prior is $0.0709$). Upper Left: $\pi = 0.0209$; Upper Right: $\pi = 0.1209$; Lower Left: $\pi = 0.1709$; Lower Right: $\pi = 0.2709$: All measures are calculated for test data sampled from the marginal distribution $p(\boldsymbol{x})$. The horizontal line is epoch of training the network, the vertical line of the upper is value of the precision and the recall.

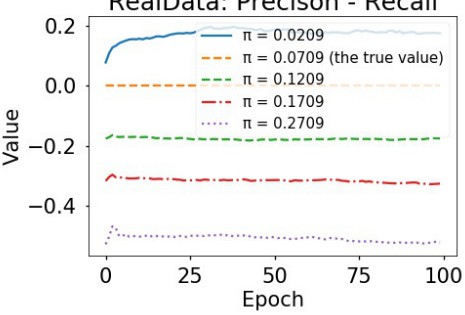

Figure 6: Experimental results of training deep neural networks using the `RealData` dataset, whose true class prior is $0.0709$, with misspecified class priors, $0.0209$, $0.1209$, $0.1709$, and $0.2709$.: All measures are calculated for test data sampled from the marginal distribution $p(\boldsymbol{x})$. The horizontal line is epoch of training the network, the vertical line of the right graph is value of the $\text{precision} - \text{recall}$.

