# OpenReview forum: "Learning from Positive and Unlabeled Data with a Selection Bias"
_ICLR.cc/2019/Conference_

### Official Review · AnonReviewer2 · 2018-11-02
**Little novelty, experiments do not offer comparison with related work**

**Rating:** 5
**Confidence:** 4

**Review:**

The authors consider the problem of learning from positive and unlabeled data in which only a subset of the true positives is labeled. While the common assumption (eg Elkan & Noto, du Plessis et al.) prescribes that the labeled set is picked independently at random from the positive set, this paper assumes that a (positive) example x is more likely to be labeled the more it exhibits positive features: formally, the higher Pr(y=1 | x), the higher Pr(o=1 | x). For instance, in the case of anomaly detection, the more likely an example is anomalous, the more likely it would get manually flagged (labeled) as positive. The authors refer to this assumption as Invariance of Order.

The proposed method requires the knowledge of the positive class prior Pr(y=1), and can be summarized in the following three steps: (i) estimate r(x)=Pr(x | y=1, o=1`)/Pr(x); (ii) find the threshold \theta such that the number of datapoints x with r(x) > \theta is a fraction Pr(y=1); (iii) train a classifier on sign(r(x) - \theta). Conceptually, the Invariance of Order assumption allows to use the order on r(x) as a proxy for an order on Pr(y=1|x), so then the knowledge of Pr(y=1) is enough to find \theta, and to port the original problem to a vanilla binary classification problem.

Concerns:
- He et al. 2018 use a very similar assumption and no comparison with that work is provided. The authors briefly mention that work in the introduction but don't perform due diligence in assessing differences/novelties with respect to that work, neither as a discussion or in the experiments.
- The requirement of knowing the fraction of positive examples is hard to justify in practice. Have you tried using the estimate obtained by Elkan et al, or other related work?
- Experiments are confusing and not convincing: apart from the very last experiment, all datasets are synthetic. No comparison with previous work is presented, except for "unbiased PU learning (PU)", which I assume is Elkan et al ? If that is indeed the case, which one of their methods are you comparing against? Even more troublesome is the fact that in all experiments you're providing your algorithm with the correct class-prior Pr(y=1), but it's not clear if this is provided to PU as well. You may want to consider estimating Pr(y=1) using methods from related work to see how it affects the accuracy.
- Related work discussion is completely missing apart from one paragraph in the introduction.

Minor:
- The acronym SCR is not very conventional; I would suggest IID which is often used as shorthand for independently identically distributed.
- Invariance of Order: when introducing it, you may want to add a sentence providing the intuition behind the assumption.
- Example 2 (Face recognition) is not very convincing and not very clear. Please rephrase.
- Pseudo-classification risk: why was the log-loss used? Can other losses be used as well?
- Theorem 3: add some intuition and explain tradeoff on \epsilon
- Experiments section: help the reader by adding a reminder on equations, as it's difficult to flip back and forth to their definitions. Eg, "we trained a classifier minimizing (4) and (7) with the model (10)" is difficult to digest and follow.
- Experiments: confusing commas in {800,1,600,3,200} => {800, 1600, 3200}
- Too many acronyms and abbreviations.

---

> ### Author Response · Authors · 2018-11-15
> **Response to AnonReviewer2**
>
> Thank you for your constructive comments. We revised our manuscript based on your advice and continue to reflect your advice on our manuscript. Our replies are listed below.
>
> Q1. He et al. 2018 use a very similar assumption and no comparison with that work is provided.
> A1.  He et al. (2018) does not seem to be a completed work and we could not discuss it. There are three reasons why a comparison with He et al. (2018) is not included in the paper.
> 1. Almost all theorems in that paper do not hold. Please see the last paragraph of this answer for examples of the incorrect points.
> 2. Although the assumption mentioned in the abstract of their paper sounds similar to ours, they added assumptions in p.7 and implicitly changed them in the proofs. These assumptions are not familiar in the context of PU learning. For example, they assume that, if p(y=+1|x) < 0.5 for a positive example, then x is not labeled. Besides, just after the declaration of the assumption, they start to use a different assumption. The actual assumption, which implicitly appears in (24), means that all unlabeled data is negative data.  Moreover, their problem setting is not the case-control scenario unlike ours.
> 3. As for experimental comparisons, it was difficult to re-implement their proposed method only from their paper. For example, they have a function beta(x) that needs to be estimated. The estimation of beta(x) requires a constrained optimization whose result depends on the optimization method. However, they did not describe what optimization method they used.  We requested the authors to send the source code before the initial submission of this paper but we did not receive a response.
>
> For these reasons, it is difficult to regard this paper as a valid reference. In the first manuscript, we mentioned this paper in the introduction to refer to the setting of their experiment. However, we deleted the paper from the related works in Introduction of the newest manuscript because we have fully confirmed that the paper is problematic as described above.
>
> Here, we describe examples of the incorrect points in that paper.  They use \rho_+ = P(\tilde{Y} = -1|X, Y=+1) in (8) and \rho_- = P(\tilde{Y} = +1|X, Y=+1) in (9) (these definitions are those from p.6, and different definitions are used in p.1). Then, they assume that \rho_+ > 0 if p(y=+1|x) > 0.5 in (16); \rho_-  = 0 if p(y=+1|x) < 0.5 in (17). As explained above, (17) is not familiar to PU learning. Furthermore, from (24) in p.9 they started to assume \rho_+  = 0  (not \rho_-=0) if p(y=+1|x) < 0.5 and proved their theorems. Combined the other assumption in (18), we must assume that all unlabeled data is negative data in order for main theorems to hold. Besides, there are also mistakes in the declaration of (16) and (17).
>
> Q2.  Have you tried using the estimate for the class prior...? Consider estimating Pr(y=1) using existing methods....
> A2.   In the initial submission we did not estimate the class prior since the prior work is not justified under our problem setting. In the newest manuscript, we added an experimental result where Ramaswamy et al. (2016) is naively used for estimation of the class prior, which is considered to be the state-of-the-art method in the “case-control” scenario with the selected completely at random assumption. In addition, we added another experiment which measures the sensitivity of our algorithm to a misspecified class prior. All results are reported in Section 5.4.
>
> Q3. Related work discussion is completely missing.
> A3. Because this paper deals with a novel problem setting, there is no directly comparable work. As explained in A1, the content of He et al. (2018) is too problematic to discuss and they discussed a different problem (not the case-control scenario). Based on the references suggested by Reviewer 1, we added a discussion regarding learning from instance-dependent label noise.
>
> Q4. The acronym SCR is not very conventional; I would suggest IID.
> A4. The term “selected completely at random” has been conventionally used since Elkan and Noto (2008). Bekker and Davis (2018) used “SCAR” to represent it, so we follow them to change the acronym to SCAR in the newest manuscript. We cannot use “IID” because our positive data is also sampled from p(x|y=+1, o=+1) in an i.i.d. manner.
>
> Q5. Why was the log-loss used?
> A5. The reason why we did not discuss other loss is that, only for the log loss function, we proved that the risk minimizer matches the density ratio that we want to estimate. Although we might use some other loss functions to approximate the conditional probability, the stationary points for these losses do not have this property in general.
>
> Q6. Theorem 3: add some intuition and explain tradeoff on \epsilon.
> A6. There is no tradeoff on \epsilon. Its value can be arbitrarily specified in the range of (0, 1/2). It is an artifact for the theoretical analysis to avoid a potential ill-definedness due to (\infty - \infty) in the problem formulation.

---

### Official Review · AnonReviewer1 · 2018-11-03
**Reasonable but somewhat unsurprising approach to an interesting problem**

**Rating:** 6
**Confidence:** 4

**Review:**

The paper proposes an approach to learning from positive and unlabelled data with a sample selection bias. Specifically, it is assumed that the observed positive instances are not necessarily drawn iid from the true positive distribution: rather, there is some bias as to which positive examples are selected. Under an assumption on the selection probability being proportional to the true probability, it is established that one may equally rank instances based on their probability of being labelled. Two algorithms are proposed for this task.

Learning under sample selection bias is an important and interesting problem. It is also arguably more realistic than the classic PU learning setting. The paper proposes a reasonable solution, which builds on some recent advances in the literature on PU learning.

My only critique is that the results are somewhat unsurprising in light of existing work on this topic (the idea of constructing unbiased risk estimators), and also on the topic of learning from label loans. Further, I believe some clarifications would better position the contributions of the paper, both in terms of strengths and limitations. More specifically:

- it seems the problem could be cast as a (interesting) special case of learning from instance dependent label noise. The assumption of the selection (i.e., label flip) probability preserving the ordering of the true class probability has a fair amount of precedent in these works; see, e.g.,

Bylander '97, Learning probabilistically consistent linear threshold functions
Du and Cai '15, Modelling class noise with symmetric and asymmetric distributions
Bootkrajang '16, A generalised label noise model for classification in the presence of annotation errors

It is in light of these works that I do not find Theorem 1 surprising. I note that the sample-selection bias setting could be seen as an interesting special case, but some discussion on the connection seems prudent.

- like in the above works, the proposed approach does not construct an unbiased estimator to the underlying risk. Instead, what is shown in e.g. Theorem 3 is that the Bayes-optimal solution to the risk is sensible. This is of course a minimal desiderata for any learning method, but unlike approaches for the classic PU learning setting, the lack of unbiasedness implies that minimizing over a restricted function class F may result in quite different solutions than if we had access to the true labels. Again, this isn't a limitation unique to this particular work, but I did feel the point could be made a little more explicit.

- also like the above work, there isn't a clear way of estimating P(y = 1). As this is crucial for the final risk estimate, it somewhat restricts the universality of the approach.

- with regards to the two algorithms proposed, both go about estimating the underlying "noisy" class probability (i.e., the probability of an instance being selected for labeling), just with different losses. While the logistic or "LSIF" loss are certainly valid choices, one could use any number of other similar loss (e.g., the exponential loss from class-probability estimation, or the "KLIEP" loss from density ratio estimation). Of course the specific choice of LSIF e.g. can be motivated since it has a closed-form solution, but the basic point is that the two approaches really boil down to changing the underlying loss function. This point could also be clarified.


Other comments:

- I believe the Elkan & Noto paper operates in the censoring rather than case-controlled setting.

- there are a few grammatical issues: e.g.., "Several recent researches", "is to find anomaly data"

- I don't follow how the case-controlled setting is "more general" than the censoring setting, as claimed in Sec 2; do you mean it is more practically realistic?

- it is correct to say in 2.1 that one cannot estimate p(y = 1 | x) from only PU data without assumptions. The next sentence states that a typical assumption that is thus made is SCR. However, this also does not guarantee that we can estimate the probability, since estimating p(y = 1) is also not possible without even further assumptions (see e.g. the mutually contaminated distribution work of Scott et al., 2013).

- in Defn 1, it would be clearer to explicate the dependence of all quantities on r.

- it is interesting that one achieves the BEP with the choice of threshold given by (2). But given that p(y = 1) is in general hard to estimate, it seems one could equally cast the problem of estimating p(y = 1) as the problem of choosing a good threshold? (This of course ignores the fact that we ostensibly need p(y = 1) when constructing the risk estimate.)

- restricting attention to scorers with output in [eps, 1 - eps] is a little strange. I assume this is in order to avoid solutions at +- infinity, which is a well-known problem with the logistic loss. It may be more natural to simply state that you operate with the extended real numbers.

- in the proof of Thm 3, I don't see the need to go through an infinite dimensional Lagrangian route. Since one is optimizing over all possible measurable functions, can one not (under suitable regularity conditions on the distribution & loss) simply compute the minimizer point wise for each x? This optimization would be a one-dimensional problem over predictions the domain [eps, 1 - eps]. The "inner risk" to be optimized (in the sense of Steinwart '06, "How to compare different loss functions and their risks") would I believe be a convex function, admitting exactly the minimizer claimed in the statement of the theorem.

- it is a bit confusing to move from F to \hat{F} as the function class.

---

> ### Author Response · Authors · 2018-11-15
> **Response to AnonReviewer1**
>
> Thank you for your insightful comments. The references you suggested are very helpful and we have included them in the newest manuscript. Our replies are listed below.
>
> Q1. It seems the problem could be cast as an (interesting) special case of learning from instance dependent label noise. The assumption of the selection probability preserving the ordering of the true class probability has a fair amount of precedent in these work. Some discussion on the connection seems prudent.
> A1. Thank you for pointing out the important related work. We have clarified the novelty of our problem in the newest manuscript. As pointed out, our problem setting can be interpreted as the case-control PU learning with instance-dependent selection bias. To the best of our knowledge, this problem has not been tackled until now either in the PU learning or the learning from label noise. In order to apply methods of learning from noisy label to PU learning, we need to assume the censoring scenario and these methods are not applicable to our problem setting based on the case-control scenario. The censoring scenario is a special case of learning from noisy label where only negative data is contaminated. Thus, in that scenario, unlabeled data can be regarded as negative-labeled data contaminated by positive data. On the other hand, in the case-control scenario, unlabeled data is from p(x), i.e., we cannot observe p(x|o = 0). Therefore, our problem is different from the work of learning from noisy label. In addition, our method is also applicable to the censoring scenario when the invariance of order holds, because the unlabeled data of the case-control scenario can be made from positive and unlabeled data of the censoring scenario.
>
> Q2. The lack of unbiasedness implies that minimizing over a restricted function may result in different solutions. I did feel the point could be made a little more explicit.
> A2. In the newest manuscript, we made the fact that we cannot obtain the unbiased estimator of the risk more explicit. Still, we believe that the most interesting point of our paper is also in this fact. It is because our main idea is to estimate the proxy for p(y=+1|x). Although our algorithm lacks unbiasedness, we can train a justifiable classier as discussed in Theorem 1. In our paper, we showed a method that can partially identify what we want to estimate instead of estimating the Bayesian optimal classifier and p(y=+1|x).
>
> Q3. There isn't a clear way of estimating P(y = 1). It somewhat restricts the universality of the approach.
> A3. We also agree with this point. There is no justified prior work for estimating the class prior in our setting. In order to answer this question as much as possible, we added two experimental results. In the first experiment, we measured the sensitivity of our algorithm to the misspecified class prior. In the second experiment, we empirically tested our algorithm with an estimated class prior. As Reviewer 2 pointed out, we can still use the existing methods in the class prior estimation as a heuristic. We used Ramaswamy et al. (2016) to estimate the class prior, which is considered to be the state-of-the-art method in the case-control PU learning. All results are reported in Section 5.4.
>
> Q4. While the logistic or LSIF losses are valid choices, one could use other similar loss.
> A4. 1) For the density ratio estimation, we will add experiment of KLIEP. 2) The reason why we did not discuss other loss except for logistic loss is that the risk minimizer matches the desired density ratio for the log loss function. Other losses may not be guaranteed to enjoy this property.
>
> Q5. Elkan & Noto paper operates in the censoring rather than case-controlled setting.
> A5. We cited their paper because they mentioned the difference between the case-control and censoring scenario. To avoid confusion, we have removed the mentioning in the newest manuscript.
>
> Q6. One achieves the BEP with the choice of threshold. It seems one cast the problem of estimating p(y = 1) as the problem of choosing a good threshold?
> A6. We agree with your opinion. As we mentioned in A3, we add new experiments to answer the question as much as possible. The experiments show that our method still works well under the estimated class prior.
>
> Q7. Restricting attention to scorers with output in [eps, 1 - eps] is a little strange.
> A7. Considering the extended real numbers may be an option, but, even in that case, a potential \infty - \infty cannot be avoided. Thus, a treatment like introducing the epsilon would be required. Furthermore, the value of epsilon is arbitrary in (0, 0.5) and it does not limit the applicability of our theoretical analysis.
>
> Q8. In the proof of Thm 3, I don't see the need to go through an infinite dimensional Lagrangian route.
> A8. Thank you for your suggestion. As you suggested, we have simplified the proof in the newest manuscript by considering a point-wise minimization of the objective functional.

---

### Official Review · AnonReviewer3 · 2018-11-04
**New technique for positive-unlabeled learning focussing on addressing selection bias.**

**Rating:** 7
**Confidence:** 2

**Review:**

In this paper, the authors present a new technique to learn from positive and unlabeled data. Specifically they are addressing the issues that arise when the positive and unlabeled data do not come from the same distribution. The way to achieve this is to learn a scoring function which preserves -the order- of the label posteriors. In other words, the authors are not making assumptions and then learning the exact posterior of p(y|...) but rather just a function r(x) with the property that if p(y_i) < p(y_j) then r(x_i) < r(x_j).

I am not super familiar in the area but I didn't see any fundamental flaws. The approach makes sense and although I cannot judge the novelty of this paper, it is a useful tool in the PU learning toolbox addressing an arguably important problem (selection bias). Except for section 5.3, the experiments are not that interesting as they are made up artificially by the authors.

Thoughts:
- In example 1, be specific about what p(y|...) and p(o|...) are.
- In example 2, I wasn't sure what p(o|...) exactly would be.
- Assumption 1, the first sentence I understand. The "if and only if" part I don't see. Can you clarify?

---

> ### Author Response · Authors · 2018-11-15
> **Response to AnonReviewer3**
>
> Thank you for your insightful comments.
> We have updated the manuscript with more explanation of our examples and a clarified statement of Assumption 1. Our replies to the questions are listed below.
>
> Q1. In example 1, be specific about what p(y|...) and p(o|...) are.
> A1. We have added more explanation in the newest manuscript. In example 1, p(y=+1|x) is the probability that a given input x is an anomaly, while p(o=+1|x) is the probability that a given input x gets labeled as an anomaly in the dataset.
>
> Q2. In example 2, I wasn't sure what p(o|...) exactly would be.
> A2. We have added more explanation in the newest manuscript. In example 2, a positively labeled data is a picture x that is known to belong to a user, while p(o=+1|x) is the probability that the user provides the picture x as a training datum.
>
> Q3. Assumption 1, the first sentence I understand. The "if and only if" part I don't see. Can you clarify?
> A3. The first and second sentences were together supposed to mean that p(y=+1|x_i) \leq p(y=+1|x_j) is equivalent to p(o=+1|x_i) \leq p(o=+1|x_j). We are sorry for the unclarity and the typo. We have updated the manuscript to make the statement clear.

---

### Meta-Review · Area_Chair1 · 2018-12-13

**Confidence:** 4
**Recommendation:** Accept (Poster)

**Metareview:**

This manuscript proposes a new algorithm for learning from positive and unlabeled data. The motivation for this work includes cases of selection bias, where the positive label is correlated with observation. The resulting procedure is shown to learn a scoring function that preserves the class-posterior ordering, and can thus be thresholded to obtain a classifier.

The problem addressed is interesting, and the approach sounds reasonable. The writing seems to be well done, particularly after the rebuttal when the work was better placed in context.

The reviewers and AC note issues with the evaluation of the proposed method. In particular, the authors do not provide a sufficiently convincing empirical evaluation on real data.